# Rapid Discovery of Substances with Anticancer Potential from Marine Fungi Based on a One Strain–Many Compounds Strategy and UPLC-QTOF-MS

**DOI:** 10.3390/md21120646

**Published:** 2023-12-18

**Authors:** Yu-Ting Wu, Xiao-Na Zhao, Pei-Xi Zhang, Cui-Fang Wang, Jing Li, Xiao-Yue Wei, Jia-Qi Shi, Wang Dai, Qi Zhang, Jie-Qing Liu

**Affiliations:** 1Engineering Research Centre of Molecular Medicine of Ministry of Education, Key Laboratory of Fujian Molecular Medicine, Key Laboratory of Precision Medicine and Molecular Diagnosis of Fujian Universities, Key Laboratory of Xiamen Marine and Gene Drugs, School of Medicine, Huaqiao University, Quanzhou 361020, China; 13142319312@163.com (Y.-T.W.); 1991020129@163.com (X.-N.Z.); 21013071029@stu.hqu.edu.cn (P.-X.Z.); 2034111009@stu.hqu.edu.cn (J.L.); 2034111021@stu.hqu.edu.cn (X.-Y.W.); 2034111019@stu.hqu.edu.cn (J.-Q.S.); wangdai05@163.com (W.D.); 2026202036@stu.hqu.edu.cn (Q.Z.); 2College of Oceanology and Food Science, Quanzhou Normal University, Quanzhou 362000, China; wangcuifang@qztc.edu.cn

**Keywords:** the secondary metabolites, marine fungi, OSMAC strategy, UPLC-QTOF-MS, cancer cell inhibition rate

## Abstract

The secondary metabolites of marine fungi with rich chemical diversity and biological activity are an important and exciting target for natural product research. This study aimed to investigate the fungal community in Quanzhou Bay, Fujian, and identified 28 strains of marine fungi. A total of 28 strains of marine fungi were screened for small-scale fermentation by the OSMAC (One Strain-Many Compounds) strategy, and 77 EtOAc crude extracts were obtained and assayed for cancer cell inhibition rate. A total of six strains of marine fungi (P-WZ-2, P-WZ-3-2, P-WZ-4, P-WZ-5, P56, and P341) with significant changes in cancer cell inhibition induced by the OSMAC strategy were analysed by UPLC-QTOF-MS. The ACD/MS Structure ID Suite software was used to predict the possible structures with inhibitory effects on cancer cells. A total of 23 compounds were identified, of which 10 compounds have been reported to have potential anticancer activity or cytotoxicity. In this study, the OSMAC strategy was combined with an untargeted metabolomics approach based on UPLC-QTOF-MS to efficiently analyse the effect of changes in culture conditions on anticancer potentials and to rapidly find active substances that inhibit cancer cell growth.

## 1. Introduction

As we all know, cancer remains a major threat to human health and is one of the deadliest diseases in the world [1]. In recent decades, the global incidence of cancer has increased dramatically, and more and more scientists have turned to natural medicines in the search for potential anticancer drugs. According to research reports, approximately 70% of more than 120 anticancer drugs are derived from natural products [2,3]. Natural products play an important role in health care and agricultural production, particularly natural products of microbial origin [4]. Antitumor drugs from microorganisms include actinomycin D, daunorubicin, adriamycin, the mitosane mitomycin C, mithramycin, streptozotocin, and pentostatin [5].

In the course of microbial research, it has been found that most synthetic gene clusters in microorganisms remain silent under conventional culture conditions, leading to a decrease in the diversity of secondary metabolites [6,7]. The “OSMAC (One Strain Many Compounds) strategy”, as a microbial culture method, has been shown to be a way to activate silent biosynthetic gene clusters in microorganisms by altering culture parameters such as medium composition, pH and culture conditions to produce more novel and high-yielding natural products [8,9,10,11,12]. The high repetitive discovery rate of secondary metabolites from microorganisms grown under single conditions, which is common in the course of research, makes it less likely that researchers will discover new compounds. The OSMAC strategy has a shorter cycle time, is more convenient and cheaper than gene editing, e.g., knocking out, introducing and xenoexpressing microbial genes and modulating promoters [6,7,13]. It has been reported that the OSMAC strategy was used to increase the yield of griseofulvin in *Hypoxylon* sp. MPO620, and studies on *Aspergillus* sp. Scsio 41501 resulted in the isolation of three new cyclohexenone derivatives, aspergispones A–C, and five new cyclohexenone derivatives, aspergispones D-H [14,15]. Currently, one of the difficulties encountered in natural product research is the repetitive discovery of known compounds. Previously, the GNPS database platform was used to construct molecular networks and genomics to predict the structure of natural products for the purpose of de-duplication [16,17,18,19]. However, it has become a challenge to rapidly detect and identify active compounds. Previously, activity-guided fractionation has been used to rapidly search for active compounds, but this traditional method suffers from the disadvantages of a high workload, low efficiency, and high cost [20]. Nowadays, non-targeted metabolomics analyses using LC-MS/MS (liquid chromatography–tandem mass spectrometry) and the ACD/MS Structure ID Suite software can rapidly find and identify active Compounds in complex samples [21,22,23,24].

In recent years, a large number of novel metabolites with a pharmacological potential have been identified from marine microorganisms, especially marine fungi [25,26,27]. In this study, marine fungi from Quanzhou Bay were isolated, purified and characterised. The OSMAC strategy was used to stimulate the strain to produce more or new active natural products that inhibit cancer cell growth. The relationship between the anticancer activity induced by the OSMAC strategy and the secondary metabolites of the marine fungi was analysed by UPLC-QTOF-MS to rapidly find the active compounds that inhibit cancer cell growth, and the possible structures with inhibitory effects on cancer cells were identified using the ACD/MS Structure ID Suite software. Our study provides a material and theoretical foundation for the discovery of potential anticancer natural products.

## 2. Results

### 2.1. Isolation and Identification of Marine Fungi

Twenty-eight strains of marine fungi were isolated from mangrove forests and along the coast in Quanzhou Bay of Fujian (Figure 1) and were identified by ITS sequence, and a phylogenetic tree was constructed (Figure 2). According to the ITS sequence alignment results (Appendix A) and morphological characteristics, the twenty-eight strains belonged to 14 fungal genera, including *Fusarium* sp., *Trichoderma* sp., *Aspergillus* sp., *Curvularia* sp., *Pestalotiopsis* sp., *Arthrinium* sp., *Rhodotorula* sp., *Nigrospora* sp., *Talaromyces* sp., *Alternaria* sp., *Neopestalotiopsis* sp., *Lasiodiplodia* sp., *Penicillium* sp., and *Diaporthe* sp.

### 2.2. Using the OSMAC Strategy to Cultivate Marine Fungi and Their Cancer Cell Inhibition Potential

Five media were selected for the culture of marine fungi using the OSMAC strategy, namely, potato dextrose broth medium (PDB), Czapek-Dox medium (Cza), Eugon medium (Eugon), malt extract medium (ME), tryptone soya medium (TS), and D-mannitol salt medium (DM). Three culture conditions were selected, namely, PDB static culture (PDB-S), PDB dynamic culture (PDB-D and PDB-attached wood culture (PDB-W). A total of 77 crude extracts of the strains were obtained using these different culture conditions and tested for their inhibition of Hela cells. The results are shown in Table 1.

We found that crude extracts from the same strain under different culture conditions had different inhibitory effects on the growth of cancer cells (Hela cells). Using the crude extracts cultured under PDB-S culture conditions as a control, six strains of marine fungi (P-WZ-2, P-WZ-3-2, P-WZ-4, P-WZ-5, P56, and P341) showed significant changes in the growth inhibition rate of Hela cells under the PDB-D or Caz culture conditions, as shown in Figure 3, and their inhibitory effects on Hela, MCF-7, A549 and HK-2 IC_50_ values and results are shown in Table 2.

### 2.3. Analysis of Differential Composition of Secondary Metabolites of Strains by UPLC-QTOF-MS

The crude extracts of the above six marine fungal strains were analysed by UPLC-QTOF-MS. The base peak chromatograms (BPCs) are shown in Figure 4A and a comparison of the peak areas of the chromatograms is shown in Figure 4B. It can be seen that there are some differential peaks in the different crude extracts of the same fungus. The differential peaks were analysed using the ACD/MS Structure ID Suite software and a total of 23 compounds were identified as shown in Figure 5. The spectral information of the identified chemical compounds **1**–**23** including retention time, molecular formula, mass (*m*/*z*), fragment ions, and anticancer activity is shown in Appendix A. Possible fragmentation pathways of the structure of compounds **1**–**23** are shown in Appendix A.

From the chromatogram of the strain P-WZ-2 (Figure 4A1), it can be seen that the five labelled peaks (A/B/C/D/E) were significantly different, and five compounds **1**–**5**, were identified. Compound **1** (Rt = 9.169 min) displayed a peak at *m*/*z* 706.3923 [M + H]^+^ and the molecular formula was deduced to be C_37_H_51_N_7_O_7_, according to the relevant literature reports, combined with the MS/MS spectrometry fragment ions, it was identified as enamidonin (**1**) [28]. Compound **2** (Rt = 14.061) had molecular ion peaks at *m*/*z* 462.2387 [M + H]^+^ and the molecular formula was deduced to be C_27_H_31_N_3_O_4_, which indicated that it was a notoamide F (**2**) [29,30]. Compound **3** (Rt = 18.0477) had molecular ion peaks at *m*/*z* 507.368 [M + H]^+^, and its molecular formula was determined to be C_30_H_50_O_6_, which was identified as penicisteroid G (**3**) [31]. Compound **4** (Rt = 24.343) and compound **5** (Rt = 25.601) were identified as oligoporin B (**4**) [32] and β-D-mannopyranoside (**5**) [33]. The peak areas in the UPLC-QTOF-MS mass spectra showed that compounds **1**–**5** were produced only under PDA-S conditions (Figure 4B1), and the content of compound **3** was significantly higher under PDA-S than under PDA-D conditions. From the chromatogram of the strain P-WZ3-2 (Figure 4A2), four differential peaks (A/B/C/D) were significantly higher than that of P-WZ3-2-1, and three compounds (**6**–**8**) were identified from the three differential peaks (A/B/D), respectively, chivosazole B (**6**) [34], trichoderolide B (**7**) [35] and cladobotric acid D (**8**) [36]. No corresponding substance for peak C was identified. The peak areas showed that the yield of compounds (**6**–**8**) under PDA-D was significantly higher than PDA-S (Figure 4B2). For the strain P-WZ-4 (Figure 4A3), seven differential peaks (A/B/C/D/E/F/G) were labelled in P-WZ4-4, peaks (A/B/C/D/E) were not observed in P-WZ4-1, and peaks (F/G) were significantly higher than that in P-WZ4-1, and a total of six compounds (**9**–**14**) were identified, namely, fragin (**9**) [37,38,39], phytosphingosine (**10**) [40,41], thielavin S (**11**) [42], penicillenol D (**12**) [43], cabanillasin (**13**) [44], and suillumide (**14**) [45]. No corresponding substance for peak D was identified. Compounds (**9**–**12**) were only produced under Cza culture medium, and the yield of compounds (**13**–**14**) under PDA-S was significantly higher than under Cza culture medium. For the strain P-WZ-5 (Figure 4A4), four differential compounds (**15**–**18**) were identified: (+)-(2S,3S,4R)-10-De-O-carbamoyl-12-O-carbamoyl-N-β-acetylstreptothricin F acid (**15**) [46]; penikellides A **16** [47,48]; (4R)-4,5-Dihydro-4-hydroxygeldanamycin (**17**) [49]; and brintonamide B (**18**) [50]. Compound **15** was only produced under PDA-W culture medium, and the yield of compounds (**16**–**18**) under PDA-W culture conditions was higher than PDA-S (Figure 4B4). For the strain P-WZ-56 (Figure 4A5), two differential compounds (**19**–**20**) were identified, preussin I (**19**) [51] and cytosporone C (**20**) [52,53], and the yield under PDA-D culture conditions was higher than PDA-S (Figure 4B5). For the strain P-341 (Figure 4A6), three compounds (**21**–**23**) were identified: 3-hydroxy-N-(1-hydroxy-3-methylpentan-2-yl)-5-oxohexanamide (**21**) [54]; eutypellone A (**22**) [55]; and CJ-13, 982 (**23**) [56]. No corresponding substance for peak B was identified. Compounds (**21**–**23**) were both produced under PDA-S and PDA-D culture media, but the yield under PDA-D culture conditions was higher than that of PDA-S (Figure 4B6).

## 3. Discussion

Previous studies have shown that the OSMAC strategy can be an effective way to activate microbial silenced biosynthetic genes and to induce strains to produce higher yields and novel secondary metabolites [57]. Therefore, in this study, we used the OSMAC strategy to culture strains and proved that the OSMAC strategy can lead to changes in the inhibition of cancer cell growth, and we hypothesised that the differences in the cancer cell inhibition rate were due to the changes in the secondary metabolites of the strains. In order to know what are the altered substances in the secondary metabolites of the strains, we identified the different substances produced by the secondary metabolites of the strains under the cultivation conditions of the OSMAC strategy using UPLC-QTOF-MS technology and the ACD/MS Structure ID Suite software, and achieved relative quantification by using the peak area and 23 compounds (**1**–**23**) were identified. A review of the literature showed that compounds enamidonin **1** [28], thielavin S **11** [42], penicillenol D **12** [43], and cabanillasin **13** [44] have antimicrobial activity. Penicisteroid G **3** [31]; chivosazole B 6 [32,33]; trichoderolide B **7** [34]; cladobotric acid D **8** [35]; phytosphingosine **10** [40,41]; suillumide **14** [45]; (4R)-4,5-dihydro-4-hydroxygeldanamycin **17** [49] have anticancer activity activity or cytotoxicity. Penicisteroid G **3** has shown moderate activity against the cell line A549 [31]. In this study, penicisteroid G **3** was identified from P-WZ-2-2, and the IC_50_ value against the A549 cell line was 14.17 ± 0.18 μM, as determined by the cancer cell inhibition rate, which is in contrast to our expected speculation about the presence of compounds in P-WZ-2-2 that inhibit cancer cell growth. Chivosazole B **6** and trichoderolide B **7** were identified in P-WZ-3-2. Chivosazole B **6** has shown activity against yeasts, filamentous fungi, and especially mammalian cells [34], and trichoderolideB **7** has been reported to have cytotoxic activity against MDA-MB-435 cells [35]. It has not been reported to inhibit cancer cell growth. However, in our study, we found that P-WZ-3-2 had inhibitory effects on MCF-7 and A549 cell lines, with IC50 values of 23.23 ± 0.22 and 38.28 ± 0.80 μM, respectively. Cladobotric acid D **8** was also identified from P-WZ-3-2-2 and has been reported to have significant cytotoxicity against the P388 cell line. Therefore, these results are in line with the expectation that the compounds inhibiting cancer cell growth we inferred in P-WZ-3-2-2 are hivosazole B **6**, trichoderolide B **7**, and cladobotric acid D **8**. Phytosphingosine **10** and suillumide **14** were identified from P-WZ-4-4. The IC_50_ value of P-WZ-4-4 against Hela, MCF-7, and A549 were 23.88 ± 0.53, 14.91 ± 0.98 and 35.88 ± 0.36 μM, respectively, but no reports were found on the inhibition rate of Phytosphingosine **10** and suillumide **14** against Hela, MCF-7, and A549. It has been reported that only phytosphingosine **10** induces mitochondria-mediated apoptosis in murine ES cells, and suillumide **14** has an IC_50_ of approximately 10 μM against the cell line SK-MEL-1 [45]**.** (4R)-4,5-dihydro-4-hydroxygeldanamycin **17** was identified from P-WZ-5-3 and is reported to show activity against HepG2 cells and has an IC_50_ of 10.8 μM [48,49]. In this study, P-WZ-5-3 had inhibitory effects on Hela, MCF-7, and A549, and its IC_50_ value was 25.29 ± 0.80, 15.9 ± 0.26, and 26.31 ± 0.69 μM, respectively. This result is in line with the expected speculation of the presence of substances with potential anticancer activity in P-WZ-5-3. Among these, preussinI **19** showed moderate to strong inhibitory activity against lipopolysaccharide-induced IL-6 production by THP-1 cells, with IC_50_ values ranging from 0.11–22 μM [51]. No studies related to preussinI **19** on cancer cells have been reported.

Moreover, for some of the differential peaks, no corresponding substance was identified, e.g., the peak C of strain P-WZ-3-2 (Figure 4A2), the peak B of P-341-2 (Figure 4A6), and the peak D of P-WZ-4-4 (Figure 4A3), which may be known compounds not included in the Natural Products Atlas database. However, this may also be due to the production of new secondary metabolites that inhibit cancer cell growth by the OSMAC strategy of culturing marine fungi, and this discovery has provided us with a follow-up study to search for new substances with potential anticancer activity. It is worth mentioning that the differential peaks with increased peak areas, such as the labelled peak C in the BPC chromatogram of P-WZ2-1 and the labelled peaks (B/C) in P-WZ3-2-2, which may be due to the increase in the production of secondary metabolites, which provides a certain research basis for subsequent largescale production and applications.

The OSMAC strategy has been shown to be a way to activate many silent biosynthetic gene clusters in microorganisms to produce more and higher yields of natural products [13,14,15,16,17]. We have analysed the active change components of marine fungi under the OSMAC strategy by the UPLC-QTOF-MS technique and rapidly identified the compounds with possible growth inhibitions of cancer cells using the Version 2022.1 of ACD/MS Structure ID Suite software. The technical roadmap for the rapid discovery of substances with potential anticancer activity in this study is shown in Figure 6. Our study provides a theoretical basis for the successful isolation of secondary metabolites with potential anticancer activity or specific structure types and also demonstrates that the efficiency of discovering potential anticancer active substances can be improved by using UPLC-QTOF-MS technology and the ACD/MS Structure ID Suite software.

## 4. Materials and Methods

### 4.1. General Experimental Procedure

EtOAc and methanol were purchased from Xilong Chemical Reagent Co., Ltd., Xiamen, China. UPLC-grade methanol and acetonitrile and formic acid were purchased from Macklin Biochemical Co., Ltd., Shanghai, China. The potato extract, glucose, agar, tryptone, and soytone were purchased from Sinopharm Chemical Reagent Co., Ltd., Shanghai, China. The D-mannitol, beef extract powder, and malt extract were purchased from Quanlong Bio-technology Co., Ltd., Nanjing, China. NaCl, NaNO_3_, KCl, K_2_HPO_4_, MgSO_4_, FeSO_4_, and L-cystine, etc., were purchased from Ocean Chemical Group Co., Ltd., Qingdao, China. The BGI 2xSuper PCR Mix (with dye) and BGI D2000 Plus DNA Ladder were purchased from Liuhe Huada Gene Technology Co., Ltd., Beijing, China. The Fungal Genome Extraction Kit was purchased from Tiangen Bio-chemical Technology Co., Ltd, Beijing, China. The agarose was purchased from Sunma Bio-technology Co., Ltd., Xiamen, China.

### 4.2. Marine Fungi Material

Samples were collected from the soil (10 cm from the surface) at Quanzhou Binhai Park (24.86 N, 118.69 E), from some from plants (seaweed and tung grass) at Quanzhou Fengche Island (24.97 N, 119.03 E), and from some from water at Quanzhou Luoyang Bridge (24.95 N, 118.68 E). All plant samples were disinfected with 75% ethanol, cut into approximately 1 mm × 1 mm fragments, and ground with a mortar. The culture media were PDB (potato extracts, 10 g; glucose, 10 g; artificial seawater, 1000 mL; pH, 7.0–7.5); Cza (glucose, 15 g; NaCl, 30 g; NaNO_3_, 1 g; KCl, 0.25 g; K_2_HPO_4_, 1 g; MgSO_4_, 0.25 g; FeSO_4_, 0.005 g; artificial seawater, 1000 mL; pH, 7.0–7.5); Eugon (tryptone, 7.5 g; soy peptone, 2.5 g; glucose, 2.5 g; L-cystine, 0.1 g; NaCl, 2 g; NaSO_3_, 0.1 g; artificial seawater, 1000 mL; pH, 7.0–7.5); ME (malt extracts, 15 g; soy peptone, 1.5 g; artificial seawater, 1000 mL; pH, 7.0–7.5); TS (tryptone, 7.5 g; soy peptone, 2.5 g; NaCl, 2.5 g; artificial seawater, 1000 mL; pH, 7.0–7.5); and DM (D-mannitol, 5 g; peptone, 5 g; beef extract powder, 1 g; NaCl, 37.5 g; artificial seawater, 1000 mL; pH, 7.0–7.5). The samples were diluted 10 1 to 10 4 times, inoculated 100 μL into the above media, spread evenly, and incubate inverted for 3 days at 28 °C. Fungal colonies were transferred to new plates and incubated until pure cultures were obtained. For cryopreservation, the single colony strain was transferred to a freezer tube and stored in a refrigerator at −20 °C.

Fungal communities were identified on the basis of the intraribosomal internal transcribed spacer (ITS) region containing the ITS1 and ITS4 regions. PCR amplification was performed using primers ITS1 and ITS4. The primer sequences were TCCGTAGGTGAACCTGCGG and TCCTCCGCTTATTGATATGC, and the fragment size was 400~800 bp. Amplification was performed in the GeneAmp 9700 Thermal Cycler using the following protocol: Initial denaturation at 96 °C for 5 min, followed by 35 cycles of DNA denaturation at 96 °C for 20 s, primer annealing at 56 °C for 20 s, and an extension at 72 °C for 10 min. A final extension step of 10 min at 16 °C completed the PCR. Sequences were compared to the NCBI Genbank (https://www.ncbi.nlm.nih.gov/genbank/, accessed on 12 March 2022) using the Nucleotide BLAST function. Fungal genera were identified according to sequence alignment results and morphological characteristics. The sequences were deposited in GenBank, and the accession numbers are given in Table 1.

### 4.3. OSMAC Strategy

The OSMAC strategy used six liquid media: PDB, Cza, Eugon, ME, TS, and DM, and three culture states were selected: PDB static culture (PDB-S); PDB dynamic culture (PDB-D, shaker at 120 rpm); and PDB woodchip culture (PDB-W, 10 g of woodchips must be added). All liquid media are prepared at 500 mL in a 1000 mL Erlenmeyer flask. All media were autoclaved at 121 °C for 20 min prior to inoculation. After inoculation, the strains were incubated at 28 °C for 28 days. The fermented fungal strains were then extracted three times with 500 mL of EtOAc together with their fermentation medium, and the EtOAc was evaporated to dryness by rotary evaporation to obtain the crude extract.

### 4.4. Cancer Cell Inhibition Rate Test

The anticancer activity of the crude extract was evaluated against the Hela cancer cell line, the MCF-7 cancer cell line, the A549 cancer cell line, and the toxicity against the non-cancerous human cell line HK-2. The CCK-8 assay was performed according to assays described in the literature. Cells were cultured in an RPMI1640 medium supplemented with a 10% fetal bovine serum and 1% double antibody. Cells are taken at the logarithmic growth phase, diluted to 5 × 10^4^/mL, and plated at 100 μL per well in a 96-well plate. After the cells grew for 24 h, the medium was discarded and DMSO was added to dissolve. The crude extract diluted with the culture medium was used for 48 h at a concentration of 100 ug/mL. After 48 h, the culture medium was discarded and the CCK-8 solution was added according to the indicated dosage, followed by incubation for 30–40 min, and the OD value at 450 nm was recorded using a microplate reader. The inhibition rate of cell growth was calculated.

### 4.5. UPLC-QTOF-MS Analysis

LC-MS-MS was performed using a 6545 LC/QTOF and a 1290 Infinity LC system (Agilent Technologies Inc., Santa Clara, CA, USA) with a YMC C18 column (YMC Co., Ltd., Kyoto, Japan) YMC-Park, ODS-A, 250 × 2.1 mm, S-5 μm, 12 nm, 0.8 mL). Sample preparation: The crude extract was dissolved in methanol to 1 mg/mL. It was then filtered into HPLC vials using a 0.2 μm PTFE syringe filter. All samples were analysed by LC-MS with the mobile phase consisting of water (A) and acetonitrile (B), both containing formic acid (0.1%, *v*/*v*), and the flow rate was set at 1.0 mL min-1. The injection volume was 10 μL, and the following gradient procedure was used: 0–22 min, 10–00%, 22–19, min 100% with an automated fully dependent MS^2^ scan. Differentiation of the protonated molecules, adducts, and fragment ions was performed by the identification of [M + H]^+^. We have publicly uploaded the MS/MS data as a MassIVE archive to the GNPS as MSV000093555.

Each LC-MS dataset (RAW file format) was loaded into ACD/MS Structure ID Suite, where the IntelliXtract algorithm (IX) was used to extract all chromatographic constituents, perform peak integrations, and group the spectral features and display them in order of retention time, as well as displaying the best matching candidates in the database, including their structure and molecular formula. We also used the Auto Assign tool in MS Structure ID Suite to score compounds on a 0–1 scale by comparing their experimental MS^2^ spectra to the predicted fragments of the candidate structures, further evaluating all matches and ultimately identifying the structure that best matches the analysed data.

## 5. Conclusions

In this study, 28 strains of marine fungi were isolated, purified, and identified from the marine area of Quanzhou Bay, and the OSMAC strategy was used to ferment the 28 strains of marine fungi on a small scale to promote the production of metabolites that inhibit cancer cell growth. A total of 77 crude extracts were obtained and tested for the cancer cell inhibition rate, and six strains of marine fungi with significant differences in the Hela cell inhibition rate were screened, which included P-WZ-2, P-WZ-3-2, P-WZ-4, P-WZ-5, P56, and P341, respectively. The differential components of the secondary metabolites of the six marine fungal strains were identified by the UPLC-QTOF-MS technique and using the ACD/MS Structure ID Suite software, and a total of 23 compounds **1**–**23** were identified. We have shown that the OSMAC strategy has a positive effect on both the abundance and diversity of secondary metabolites in marine fungi and that changes in culture conditions have a direct effect on the growth rate of strains against cancer cells. And our study is of great value for the full exploitation and utilisation of marine fungal resources. Meanwhile, the application of UPLC-QTOF-MS technology has laid a material and theoretical foundation for the targeted separation of secondary metabolites with anticancer activity or specific structural types and has improved the discovery efficiency of anticancer active substances.

## Figures and Tables

**Figure 1 marinedrugs-21-00646-f001:**
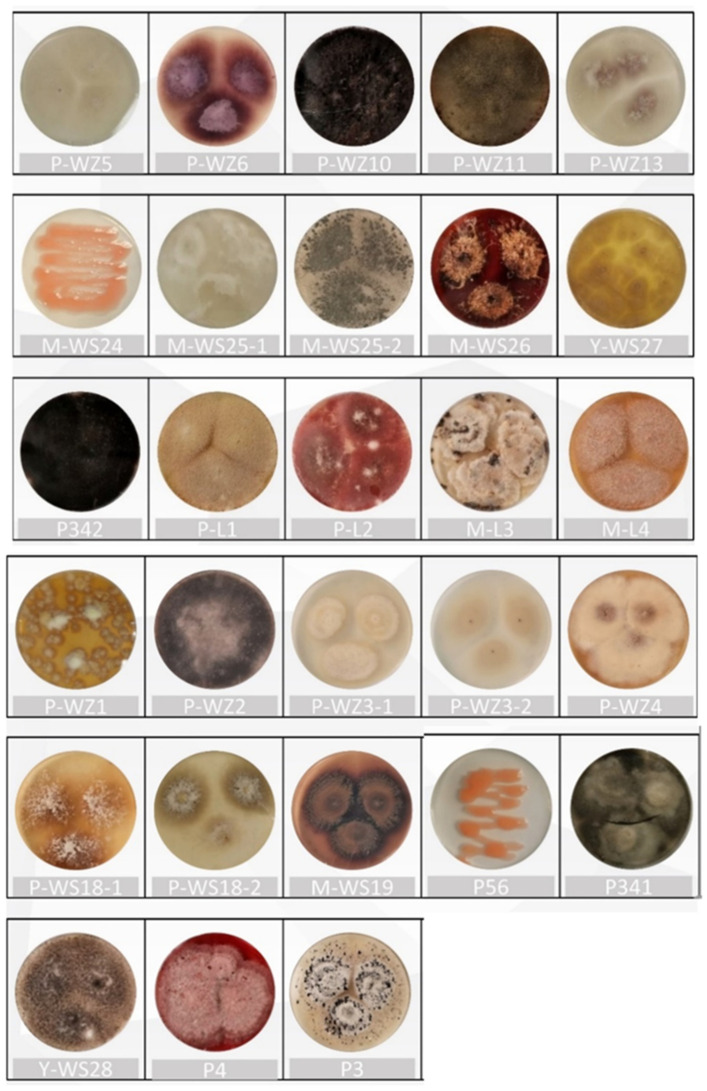
Morphological characteristics of 28 marine fungus strains on solid medium.

**Figure 2 marinedrugs-21-00646-f002:**
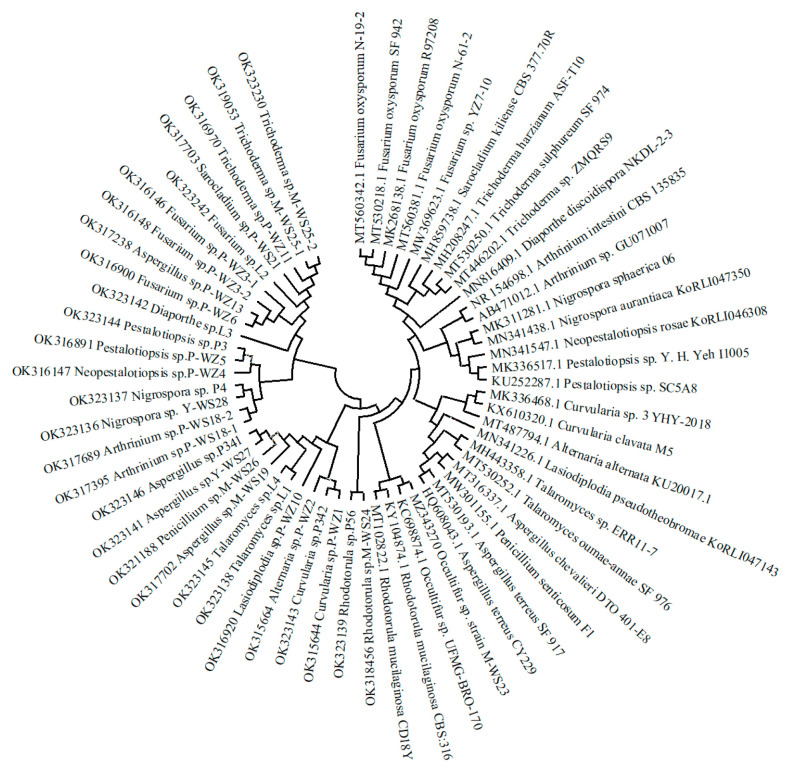
A phylogenetic tree of 28 strains of marine fungi.

**Figure 3 marinedrugs-21-00646-f003:**
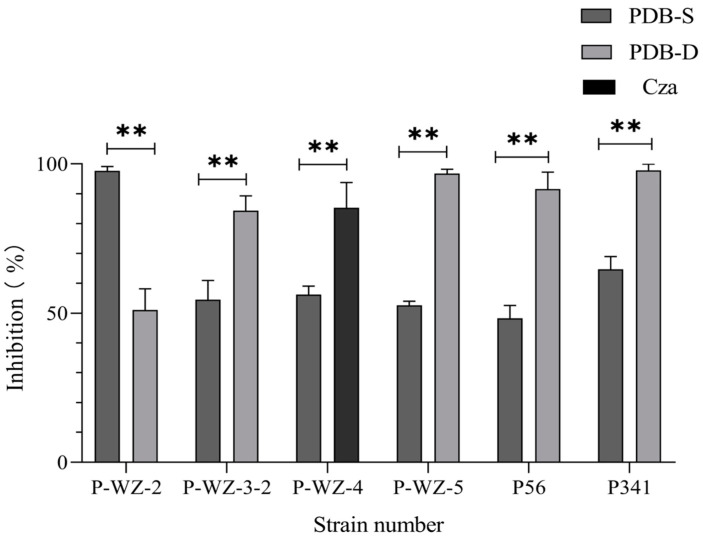
Comparison of Hela cell inhibition rates under different culture strips of 6 strains of marine fungi. ** *p* < 0.01, indicating a statistically significant difference.

**Figure 4 marinedrugs-21-00646-f004:**
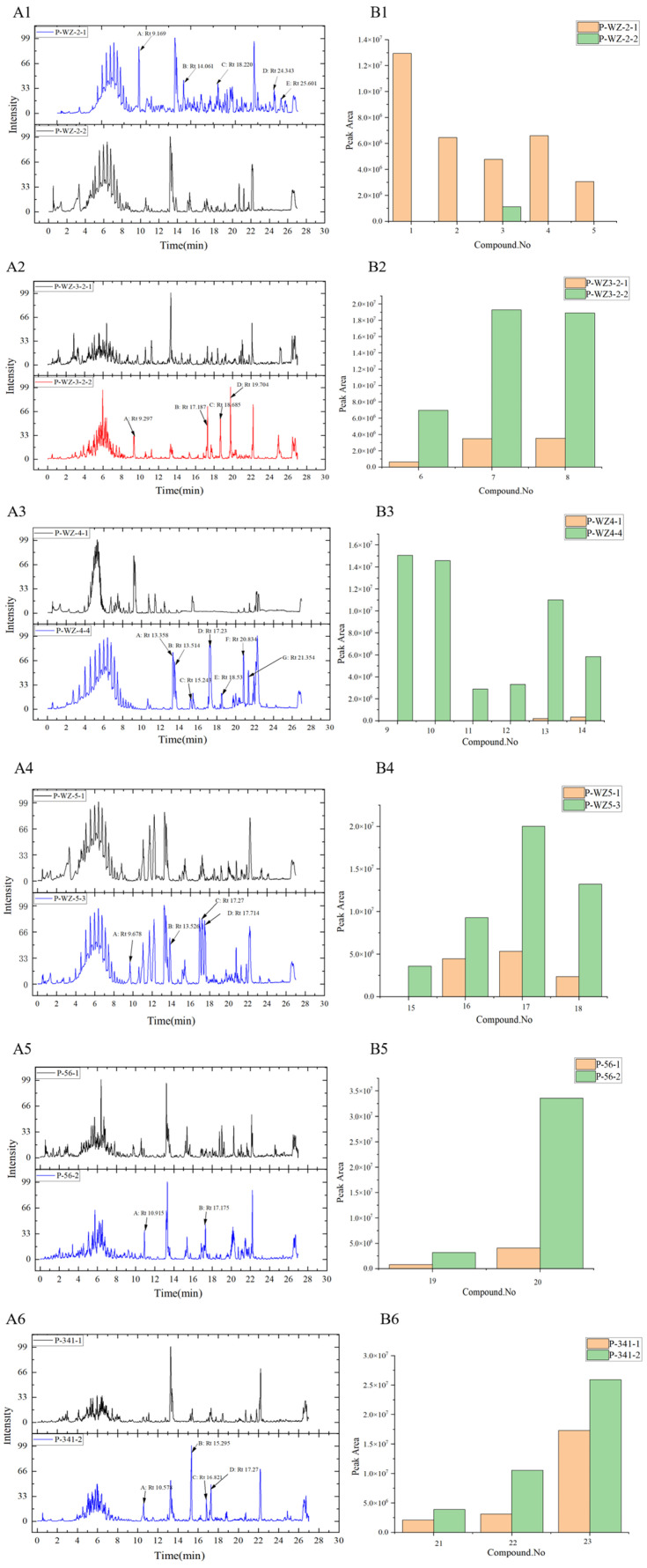
(**A**) Base peak chromatograms of strains with significant differences in cancer cell inhibition were analysed by UPLC-QTOF-MS. Rt: the peak time (min). (**B**) Comparison of the peak area of the differential compounds identified from the BPC chromatograms of the strains under different culture conditions. (**A1**) The BPC chromatogram comparison of crude extracts (P-WZ-2-1 and P-WZ-2-2) of the strain P-WZ-2 cultured in PDA-S and PDA-D, respectively. (**B1**) A peak area comparison of the chromatograms of compounds **1**–**5**, identified from the strain P-WZ-2. (**A2**) The BPC chromatogram comparison of crude extracts (P-WZ3-2-1 and P-WZ3-2-2) of the strain P-WZ3-2 cultured in PDA-S and PDA-D, respectively. (**B2**) A peak area comparison of the chromatograms of compounds **6**–**8** identified from the strain P-WZ3-2. (**A3**) A BPC chromatogram comparison of crude extracts (P-WZ-4-1 and P-WZ-4-4) of the strain P-WZ-4 cultured in PDA-S and Cza culture media, respectively. (**B3**) A peak area comparison of the chromatograms of compounds **9**–**14** identified from the strain P-WZ-4. (**A4**) The BPC chromato-gram comparison of crude extracts (P-WZ-5-1, P-WZ-5-2, and P-WZ-5-3) of the strain P-WZ-5 cultured in PDA-S, PDA-D, and PDA-W, respectively. (**B4**). A peak area comparison of the chromatograms of compounds **15**–**18**, identified from the strain P-WZ-5. (**A5**) A BPC chromatogram comparison of crude extracts (P-56-1 and P-56-2) of the strain P-56 cultured in PDA-S and PDA-D, respectively. (**B5**) A peak area comparison of the chromatograms of compounds **19**–**20**, identified from the strain P-WZ-5. (**A6**) A BPC chromatogram comparison of crude extracts (P-341-1 and P-341-2) of the strain P-341 cultured in PDA-S and PDA-D, respectively. (**B6**) A peak area comparison of the chromatograms of compounds **21**–**23**, identified from the strain P-341.

**Figure 5 marinedrugs-21-00646-f005:**
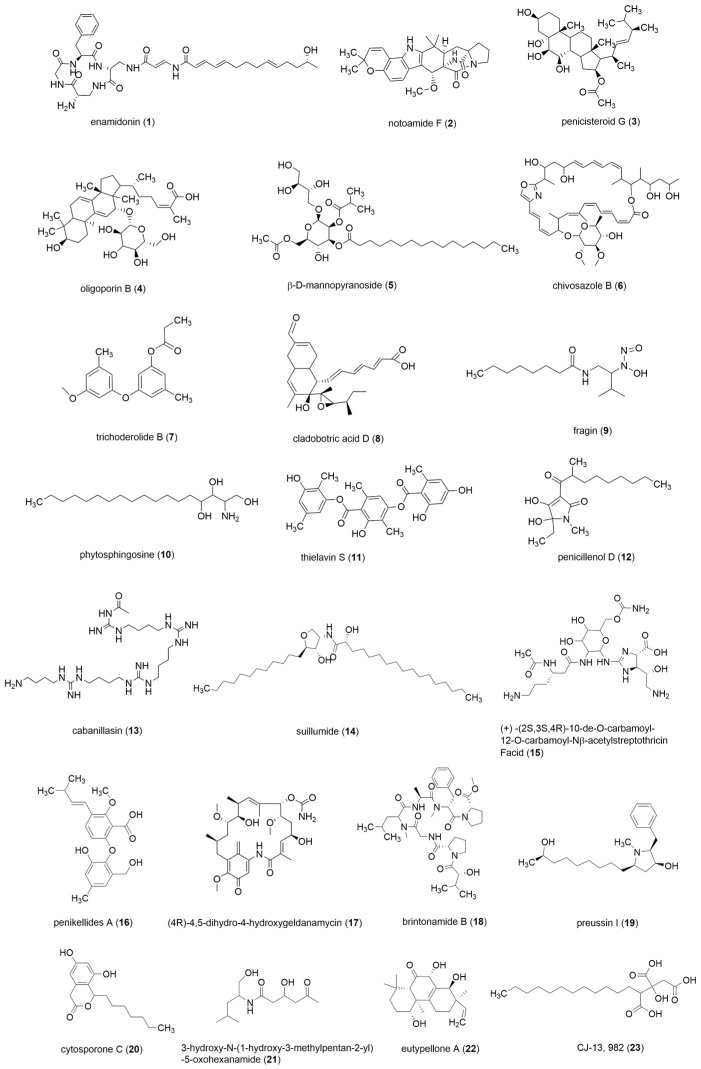
The structure of compounds **1**–**23**.

**Figure 6 marinedrugs-21-00646-f006:**
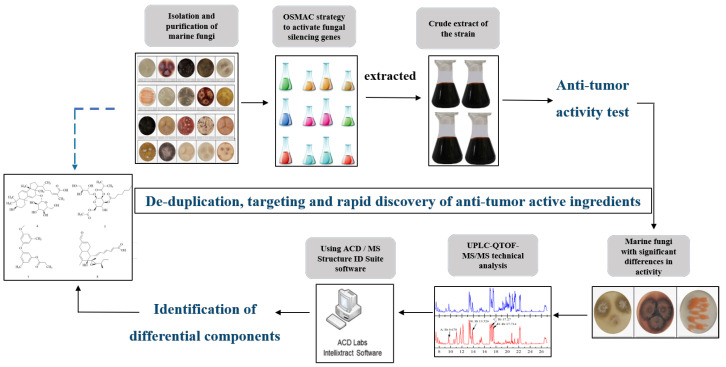
Technology roadmap for rapid discovery of substances with potential anticancer activity.

**Table 1 marinedrugs-21-00646-t001:** Inhibition of Hela cell growth by 28 strains under different culture conditions.

Crude Extracts	Culture Conditions	Growth Inhibition (%)	Crude Extracts	Culture Conditions	Growth Inhibition (%)
P3-1	PDB-S	78.07	P-WZ-4-1	PDB-S	57.23
P3-2	PDB-W	84.38	P-WZ-4-4	Cza	87.31
P3-3	PDB-D	81.71	P-WZ-4-5	Eugon	77.07
P3-5	Eugon	82.32	P-WZ-4-6	ME	62.81
P3-6	ME	75.25	P-WZ-4-7	Ts	63.71
P3-7	Ts	78.16	P-WZ-4-8	DM	75.47
P3-8	DM	91.48	P-WZ-5-1	PDB-S	53.62
P-WZ-1-1	PDB-S	96.03	P-WZ-5-2	PDB-W	97.24
P-WZ-1-2	PDB-W	87.66	P-WZ-5-3	PDB-D	96.78
P-WZ-1-3	PDB-D	78.25	P-WZ-5-4	Cza	81.30
P-WZ-1-4	Cza	89.33	P-WZ-5-5	Eugon	66.17
P-WZ-1-5	Eugon	93.03	P-WZ-5-6	ME	64.15
P-WZ-1-6	ME	73.06	P-WZ-5-7	Ts	85.12
P-WZ-1-7	Ts	69.93	P-WZ-5-8	DM	79.09
P-WZ-1-8	DM	77.62	P-WZ-6-1	PDB-S	82.76
P4-1	PDB-S	99.11	P-WZ-6-2	PDB-D	56.75
P4-2	PDB-D	99.16	P-WZ-3-1-1	PDB-S	98.00
P56-1	PDB-S	48.30	P-WZ-3-1-2	PDB-D	98.20
P56-2	PDB-D	91.62	P-WZ-3-2-1	PDB-S	55.06
P341-1	PDB-S	63.71	P-WZ-3-2-2	PDB-D	85.86
P341-2	PDB-D	99.37	P-WZ-2-1	PDB-S	97.73
P342-1	PDB-S	69.44	P-WZ-2-2	PDB-D	51.06
P342-2	PDB-W	64.13	P-WZ-10-1	PDB-S	92.32
P-WZ-13-1	PDB-S	91.47	P-WZ-10-2	PDB-D	100.00
P-WZ-13-2	PDB-D	93.84	P-WZ-11-1	PDB-S	80.83
P-WS-18-1-1	PDB-S	94.31	P-WZ-11-2	PDB-D	86.94
P-WS-18-1-2	PDB-D	70.57	M-WS-26-1	PDB-S	91.16
P-WS-18-2-1	PDB-S	88.96	M-WS-26-2	PDB-D	74.49
P-WS-18-2-2	PDB-D	97.50	M-WS-24-1	PDB-S	81.27
P-L1-1	PDB-S	93.51	M-WS-24-2	PDB-D	100.00
P-L1-2	PDB-D	93.59	M-WS-25-1-1	PDB-S	90.03
P-L2-1	PDB-S	98.62	M-WS-25-1-2	PDB-D	100.00
P-L2-2	PDB-D	99.69	M-WS-25-2-1	PDB-S	91.81
P-L3-1	PDB-S	67.16	M-WS-25-2-2	PDB-D	68.79
P-L3-2	PDB-D	78.49	M-WS-19-1	PDB-S	96.51
P-L4-1	PDB-S	86.27	M-WS-19-2	PDB-D	95.37
P-L4-2	PDB-D	92.01	Y-WS-28-1	PDB-S	87.09
Y-WS-27-1	PDB-S	97.61	Y-WS-28-2	PDB-D	55.33
Y-WS-27-2	PDB-D	98.50			

**Table 2 marinedrugs-21-00646-t002:** The IC_50_ values of six strains of marine fungi against cancer cells under different culture conditions.

Strain	Crude Extracts	IC_50_ (µM)			
		Hela	MCF-7	A549	HK-2
P-WZ-2	P-WZ-2-1	17.01 ± 0.44	20.62 ± 0.16	14.17 ± 0.18	21.6 ± 0.54
	P-WZ-2-2	63.85 ± 0.68	-	-	76.45 ± 0.47
P-WZ-3-2	P-WZ-3-2-1	-	25.98 ± 0.77	-	27.2 ± 0.28
	P-WZ-3-2-2	-	23.23 ± 0.22	38.28 ± 0.80	34.28 ± 0.91
P-WZ-4	P-WZ-4-1	76.12 ± 0.72	-	-	76.9 ± 0.45
	P-WZ-4-4	23.88 ± 0.53	14.91 ± 0.98	35.88 ± 0.36	9.569 ± 0.30
P-WZ-5	P-WZ-5-1	79.42 ± 0.96	-	72.19 ± 0.26	-
	P-WZ-5-3	25.29 ± 0.80	15.9 ± 0.26	26.31 ± 0.69	19.01 ± 0.91
P56	P56-1	85.6 ± 0.29	-	94.31 ± 0.71	-
	P56-2	25.91 ± 0.73	-	19.64 ± 0.59	49.22 ± 0.55
P341	P341-1	64.88 ± 0.56	53.22 ± 0.39	71.94 ± 0.26	43.31 ± 0.25
	P341-2	42.14 ± 0.20	34.5 ± 0.72	23.48 ± 0.23	36.66 ± 0.81

“-” Indicates IC_50_ > 100 µM. Data are from three independent experiments (*n* = 3, mean ± SD).

## Data Availability

The data used to prepare this manuscript are contained within the article and its corresponding Appendix A.

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
