# Peer review of "Rapid Discovery of Substances with Anticancer Potential from Marine Fungi Based on a One Strain–Many Compounds Strategy and UPLC-QTOF-MS"

_marinedrugs, 2023, doi:10.3390/md21120646_

Round 1

Reviewer 1 Report

Comments and Suggestions for Authors

The manuscript reports the discovery of antitumor compounds from different marine fungal strains by using an OSMAC approach. This strategy has been proved to be a simple and powerful tool to promote the production of different secondary metabolites, which represents a promising approach for the discovery of new bioactive compounds. In the present article, based on this strategy, the authors selected 5 different culture conditions for 28 marine fungal strains and they analyzed the contents and antitumor activities of 77 crude extracts. In this analysis, they identified up to 23 known metabolites, whose compositions depended on the culture conditions, and they detected different antitumor activities for the different extracts. Despite the interest and importance of the OSMA strategy for the discovery of new bioactive compounds, I consider that the present manuscripts does not properly exploit this tool. In this sense, the authors should address the following questions and issues:

1. Why have not the authors considered the use of epigenetic modifiers in the culture media to trigger new metabolic profiles?.

2. The authors should identify the compounds corresponding to the peaks C and D. To this aim, these compounds should be isolated and characterized.

3. Discussion and conclusion sections must be significantly improved, with a clear correlation and rationale between extract composition and antitumor activity.

4. In page 6, it is mentioned tables 3 and 4. Where are they?.

4. Finally, some aspects of the format must be revised: a) In Introduction section, there are some sentences that requires a grammar revision; b) the quality of the drawings of the molecular structures of Figure 5 must be improved and ordered according the numbering. I suggest to include the names of the known compounds; c) Some numbers of the compounds must be in bold; d) Revise format of the references.

In consequence, the manuscript requires major revision according to the above indications before final acceptance for publication.          

Comments on the Quality of English Language

In general, quality of English is fine but some sentences in Introduction Section requires revision. For example: "Accelerated discovery and identification....."

Author Response

“Please see the attachment”

Reviewer 2 Report

Comments and Suggestions for Authors

Attached.

Comments on the Quality of English Language

A couple issues I pointed out, but overall well written, probably worth checking again for minor wording issues and typos.

Reviewer 3 Report

Comments and Suggestions for Authors

The present paper presents results on secondary metabolites using the OSMAC strategy and UPLC-QTOF-MS/MS for the identification, identifying 28 strains of marine fungi from Quanzhou Bay. The article is well written and the results seems well documented. The OSMAC strategy is fairly novel, introduce a couple of decades ago. The authors give results showing that this may be an efficient strategy for mapping of a large number og strains and metabolites. Some interesting results regarding anti-tumor effects of the total of 23 compounds tested, is also interesting. Prior to publication, the only change in the manuscript is Figure 5. It seems very illogical to have structures 13-23 at top, and structures 1-12 at the bottom. Please put them in numerical order from the top. Other than this, I recommend publication.

Round 2

Reviewer 1 Report

Comments and Suggestions for Authors

The authors have addressed all the questions inquired in the first review. The manuscript can be accepted in present form.

Reviewer 2 Report

Comments and Suggestions for Authors

Authors have address almost every concern I had with original submission so I have only two corrections before publication:

-Authors have been diligent in changing the text to be clear that their paper describes compounds assayed for cancer cell inhibition, not in vivo antitumor activity. However, I think the revision to the title is insufficient. Cancer is the disease state in the organism, which isn't be tested for in this paper, so "Anticancer Substances" should be changed to "Cytotoxic Substances" or "Substances with Anticancer Potential" or "Substances Toxic to Cancer Cell Lines". Also, I think an article is needed in the title before OSMAC, so maybe "an OSMAC Strategy".

-Author response letter mentioned incorporation of ppm error calculations into Table S2, but I don't see that information. So it that might be an older version of the table in there that still needs to be replaced with the new version that has ppm error.

Comments on the Quality of English Language

I found a couple minor typos. In the abstract the OSMAC abbreviation should be defined. Page 3 Line 136 still doesn't look like it has a space between the capital "F" and the word "acid". Is that just a formatting issue?
